# Robotic Single-Site Radical Hysterectomy for Early Cervical Cancer: A Single Center Experience of 5 Years

**DOI:** 10.3390/jpm13050733

**Published:** 2023-04-26

**Authors:** Changho Song, Tae-Kyu Jang, Soomin Kong, Heeju Kang, Sang-Hoon Kwon, Chi-Heum Cho

**Affiliations:** Department of Obstetrics and Gynecology, Keimyung University School of Medicine, Daegu 42601, Republic of Korea

**Keywords:** cervical cancer, radical hysterectomy, robot assisted surgery, single-site

## Abstract

Background: The mainstay of treatment for early-stage cervical cancer is surgery; we present a 5-year experience of robotic single-site radical hysterectomy (RSRH) focused on surgical and oncologic outcomes. Methods: This retrospective study included 44 cases of RSRH performed in patients with early-stage cervical cancer. Results: The median follow-up period for the 44 patients was 34 months. The mean total operation time was 156.07 ± 31.77 min, while mean console time was 95.81 ± 24.95 min. Two cases had complications, which required surgical management, while four cases (9.1%) exhibited recurrence. The disease-free survival rate at 5 years was 90.9%. The sub-division analysis showed that Stage Ia2 and stage Ib1 patient sub-group showed better DFS than that of the stage Ib2 patient sub-group. The learning curve analysis showed that the CUSUM-T initially peaks at the sixth case then gradually decreases before rising and peaking at the 24th case. After 24th case, the CUSUM-T gradually decreases and reaches zero. Conclusion: The surgical outcomes of RSRH for early-stage cervical cancer treatment were safe and acceptable. However, RSRH could be considered carefully only in well-selected patient groups. Large-scale prospective studies are necessary in the future to validate the results.

## 1. Introduction

For the past few decades, most developed countries have widely accepted and utilized screening tests for cervical cancer and human papilloma virus vaccination, which has resulted in a remarkable decrease in the incidence of cervical cancer [1]. However, cervical cancer is still ranked as the fourth most common cancer in females and the fourth leading cause of death in females worldwide, including in developing countries [2].

The International Federation of Gynecology and Obstetrics (FIGO) revised the cervical cancer staging system, which was most widely used for cervical cancer staging, in 2018 [3]. The revised version incorporates an image test into the staging process, enabling more accurate staging [4]. Accurate staging is essential for planning the initial treatment strategy of a patient with cervical cancer. For a locally advanced stage of cervical cancer, radiation with or without low dose chemotherapy is the primary treatment option; chemotherapy is used in advanced stage cervical cancer [5]. The mainstay of treatment generally considered for early-stage cervical cancer below FIGO stage IIa1 is surgery [6,7]. The standard treatment for early-stage cervical cancer patients who do not wish to preserve fertility is radical hysterectomy with pelvic lymph node dissection [8,9]. Conventionally, radical hysterectomy was performed via an abdominal or laparoscopic approach [10]. However, there have been remarkable advances in minimally invasive surgery (MIS), including and single- and multi-port laparoscopic surgery. Moreover, the robotic system, which possesses the advantages of ergonomics since it automatically inverses the instrument, has emerged as an optimized MIS method utilized by some gynecologic surgeons in the treatment of cervical cancer. In this study, we present the surgical and oncological outcomes of robotic single-site radical hysterectomy (RSRH) for early-stage cervical cancer treatment based on the 5-year experience in a single center.

## 2. Materials and Methods

### 2.1. Patients

This retrospective study included 44 cases of RSRH in patients with early-stage cervical cancer, which were performed at Keimyung University Dongsan Hospital, Daegu, Korea, from 2015 to 2020. The RSRHs were performed by a single surgeon. This retrospective cohort study was approved by the Institutional Review Board of Dongsan Medical Center (IRB No. 2022-01-029). The requirement for informed consent was waived owing to the retrospective nature of the study. The indications for RSRH were patients with FIGO stage Ia2, Ib1, and Ib2 without a risk of massive adhesion owing to previous operations. The patients were informed about the result of resent randomized controlled trial before making a decision regarding the surgical approach [11]. The FIGO stage was adjusted according to the revised FIGO stage for patients who were diagnosed before 2018.

### 2.2. Surgical Methods

The robot platform used in this study was the da Vinci Si or X (Intuitive surgical, Sunnyvale, CA, USA). Under adequate general anesthesia and proper surgical drape, a vertical incision of 2.5 cm was performed in the periumbilical area and the abdominal cavity was opened using the Open Hasson procedure. After placing a silicon single-site port, the pneumoperitoneum was accessed at 12 mmHg with the CO_2_ gas insufflator. After inserting a trochar for the 8.5-mm robotic endoscope (30°), the docking procedure was performed, followed by the insertion of the robotic endoscope with the exploration of the abdominal cavity. Next, 5-mm curved trochars were inserted into each side of the silicon single-site port, followed by the insertion of semi-rigid surgical instruments. A monopolar hook was inserted into the trochar in the patients’ left side and fenestrated bipolar forceps into the opposite side. RUMI uterine manipulator (Cooper Surgical Inc. Trumbull, CT, USA) was inserted to manipulate the uterus. To prevent tumor spillage during the manipulation, specially edited gauze which was placed in KOH cup of RUMI manipulator and an endo specimen bag was used when retrieving the specimen. Radical hysterectomy was sequentially performed from the dissection of both the internal and external lymph nodes in each side of the pelvis to the type C1 hysterectomy [12]. The collected lymph nodes and resected uterus with adnexa were placed in an endo specimen retrieval bag (Medtronic, Dublin, Ireland) and removed via the colpotomy site. The colpotomy site was sutured using a unidirectional barbed suture of either V-Loc^TM^ (Medtronic, Dublin, Ireland) or Monofix (Samyang biopharma, Seoul, Republic of Korea).

### 2.3. Learning Curve Analysis

Learning curves were quantitatively assessed by performing the cumulative sum (CUSUM) method. The CUSUM is the running total of differences between the individual data points and the mean of all data points [13,14]. In this study, CUSUM of total operation time (CUSUM-T) was calculated. The inflection point of the slope of the CUSUM curve was considered as the breakpoint of the learning curve.

### 2.4. Survival Analysis and Cox Regression Analysis

Disease-free survival (DFS) was calculated from the day of surgery to the day of cancer-related death or confirmed recurrence by radiologic tests, including computed tomography, magnetic resonance image, and positron emissary tomography. Patients who were alive until the last follow-up period or who were lost to follow-up were considered censored observations. The DFS analysis was performed using the Kaplan Meier survival analysis and Cox regression analysis using a survival package (version 3.213; http://CRAN.R-project.org/package=survival, accessed on 3 February 2022) in R language (version 3.4.1; http://cran.r-progect.org/, accessed on 3 February 2022) [15]. The log-rank test was used to compare the Kaplan–Meier survival curves; log-ranks with a *p*-value of less than 0.05 was considered statistically significant. The hazard ratio (HR) of the 95% confidence interval (CI) was computed and a *p*-value less than 0.05 was considered statistically significant.

### 2.5. Statistical Methods

R language was used to analyze the data. Levene’s test was performed to analyze the equality of the variances. Student t-test was performed to analyze the difference in the continuous variables. Pearson’s chi-square and Fisher’s exact tests were conducted for the categorical variables. A *p-*value of less than 0.05 was considered statistically significant.

## 3. Results

### 3.1. Clinical Characteristics

The patient characteristics are listed in Table 1. The median age of the patients was 45 years (ranged from 30 to 65 years). The mean body mass index (BMI) was 23.96 ± 4.04. According to the classification of BMI devised by the World Health Organization, 26, 10, 4, and 4 patients were normal weight, pre-obesity, underweight, and obesity class I, respectively. Five, 27, and 12 patients were in stage Ia2, Ib1, and Ib2, respectively. In total, 33 patients were diagnosed with squamous cell carcinoma and 11 patients with adenocarcinoma.

### 3.2. Surgical Outcomes

The overall surgical outcomes of the cases are listed in Table 2. The mean total operation, docking, and console times were 156.07 ± 31.77, 6.05 ± 2.85, and 95.81 ± 24.95 min, respectively. The median count of the retrieved lymph nodes was nine, ranging from two to 20; none of them showed tumor involvement. Surgical complication was sub-categorized into two groups based on the modification of the classification of surgical complications reported in 2004 [16]. The minor complications in this study included grade I and II surgical complications, which were defined as any deviation from the normal post-operative course with or without the need for pharmacological treatment; the major complications included grade III or greater surgical complications, which were defined as complications warranting surgical intervention, life threatening complications, or even the death of a patient. There were 10 cases of minor complications, including one case of lymphocele, four cases of urinary retention, and two cases of post-operative pain. Three cases included, as a minor complication, excessive drainage exceeding 300 cc daily and which required 1 or 2 days of prolonged hospital stay. There were two major complications: vaginal cuff disruption and rectal perforation. These cases were surgically repaired. The median number of days per hospital stay was 5 days, with actual stay periods ranging from 3 to 60 days.

### 3.3. Higher BMI Is Associated with Longer Console Time and Longer Total Operation Time

In order to analyze whether the patient’s characteristics affect the surgical outcomes, we sub-divided the patients into two groups for each of the categories. Age was sub-divided into groups aged over 60 and 60 or younger. BMI was sub-divided into groups aged under 25 and 25 or over. Parity was sub-divided into a nulliparous group and a group of women who had experienced one or more births. The stages were sub-divided into groups of stages Ia2 and Ib1 and stage Ib2. The results demonstrated that the console and total operation times were longer in both groups that had patients with a higher BMI. The setting time was longer and the closure time was shorter in the nulliparous women group (Table 3). However, the sub-divisions of the patient’s characteristics other than BMI and parity affected neither the surgical nor the oncological outcomes.

### 3.4. The CUSUM Learning Curve Analysis of Robot Assisted Single-Site Radical Hysterectomy

The cumulative average of the total operation time is illustrated in Figure 1. The logarithm curve was the optimal fitting model. The equation, *p-*value, and optimization level r^2^ were as follows: cumulative average of the total operation time (min) = −11.11lnx + 175.54, *p* = 9.9 × 10^−13^, and r^2^ = 0.706. The curve demonstrates the gradual decrease in the cumulative average of the total operation time. The CUSUM learning curve analysis demonstrated three phases in the learning curve (Figure 2). The cubic curve was the optimal fitting model of CUSUM-T. The equation, *p-*value, and optimization level r^2^ were as follows: CUSUM-T (min) = −0.016x^3^ + 0.781x^2^ − 7.225x + 318.8, *p* = 1.54 × 10^−6^, and r^2^ = 0.5209. In phase 1, the CUSUM-T rose and peaked at the sixth case. The mean total operation time in phase 1 was 197.4 ± 29.0 min. Phase 2 was maintained via the subsequent 18 cases, which were represented by the positive slope. The mean total operation time in phase 2 was 158.5 ± 29.7 min. Phase 3 was represented through the negative slope as CUSUM-T decreased and reached zero. The mean total operation time in phase 3 was 139.6 ± 21.3 min. The learning curve demonstrated that the CUSUM-T initially peaks at the sixth case, gradually decreases, and then rises and peaks at the 24th case. After the 24th case, the CUSUM-T gradually decreases and reaches zero. There were no significant differences between the former six cases and the latter 38 cases based on the age group, BMI group, parity group, and stage group; however, the console and total operation times were significantly different between the two phases (129.8 ± 19.1 vs. 89.8 ± 21.2 min in console time, respectively; *p* = 0.00008 and 197.7 ± 29.0 vs. 148.6 ± 27.0 min in total operation time, respectively; *p* = 0.00001). Interestingly, no significant difference was observed between the first 24 cases and the latter 20 cases based on the age, BMI, parity, and stage groups; however, the closure, console, docking, and total operation times were significantly shorter for the latter 20 cases than the first 24 cases (Table 4).

### 3.5. Oncological Outcomes

The overall oncological outcomes are listed in Table 5. The decision making for adjuvant therapy following surgery was made in the tumor board meeting according to the National Comprehensive Cancer Network (NCCN) guidelines [7,17]. In the tumor board meeting, post-operative information regarding intermediate risk-factors, including tumor size, lympho-vascular space invasion status and stomal invasion depth, were carefully reviewed specialists in cancer. The patients who have two or more intermediate risk factors were encouraged to undergo either radiation therapy or concurrent chemo-radiation therapy (CCRT) [18]. There were two cases with resection margin involved by tumor. There was no upstaged case after surgery by tumor size and lymph node involvement, but there was three cases with positive parametrial invasion which were not found in pre-operative imaging. There was no case of tumor size exceeding 4 cm. Two patients underwent only radiation therapy and 13 patients underwent concurrent chemo-radiation therapy (CCRT). The median follow-up period for 44 patients was 34 months. Four cases (9.1%) of recurrence with mean recurrence times of 16.9 months were present. Two cases had recurred in the lungs, while two cases recurred in the vaginal stump. Three out of the four recurred cases underwent adjuvant CCRT; however, one case, which recurred in vaginal stump, did not undergo adjuvant therapy. The histological sub-types of the recurred cases were squamous cell carcinomas with the exception of 1 case, which was of the mucinous adenocarcinoma sub-type. The DFS at 5 years was 90.9% (Figure 3). Only one case of cancer-related death was observed which had no risk factor for recurrence after surgery. The survival analyses were conducted through sub-dividing the patient’s characteristics into age, BMI, parity, and stage. It demonstrated that the sub-division of the patient’s characteristics of age, BMI, and parity did not affect the DFS of the patients with cervical cancer. The sub-divisions of stage Ia and Ib patient sub-group did not show any significant difference in the DFS. In contrast, the stage Ia2 and stage Ib1 patient sub-group showed better DFS than the stage Ib2 patient sub-group (Figure 4).

## 4. Discussion

The standard surgical treatment for early stage cervical cancer is radical hysterectomy. It can be performed either through conventional abdominal surgery or MIS, including robotic surgery [19,20]. In this study, the data of the 44 patients with early stage cervical cancer who had undergone RSRH were analyzed, focusing on the surgical and oncological outcomes.

The mean total operation and console time were 156.07 ± 31.77 and 95.81 ± 24.95 min, respectively. Considering the absence of cases requiring transfusion and conversion to laparoscopy or laparotomy, the surgical outcome seems rather favorable. Although 12 cases showed complications, only two required proper surgical management following operation. One case was rectal perforation. The patient had abdominal pain and fever 2 days after the surgery. Primary closure was performed in perforated rectum by the colorectal surgical team. The other case of complication was vaginal cuff disruption. The patient visited the emergency room 7 days after discharging from hospital due to vaginal spotting and vaginal discharge. Vaginal cuff disruption was confirmed and vaginal cuff repair was performed.

The learning curve of a certain surgery is defined as the requirement of the cases until the operation time and surgical outcomes are stabilized [21]. CUSUM analysis was originally developed for use in quality control in the industrial process; however, nowadays it is widely used in the medical field to estimate the least number of cases required for acquiring stable skills for a new procedure [14]. In this study, CUSUM analysis was implemented to analyze the total operation time of robot-assisted single-site radical hysterectomy. The total operation time comprises the setting, preparation, docking, console, and closure time. We focused on the total operation time because it represents not only the learning curve of the operator but also that of the entire surgical team. CUSUM-T chart showed two peaks and three phases. The initial peak and an inflection point was reached at the sixth case. The mean console time and the mean total operation time were significantly longer in the first six cases than in the latter 38 cases, while the patient’s characteristics were not significantly different. This finding suggests that the operator in our study, who was already familiar with robotic surgery, required at least six cases to acquire basic skills for robot-assisted single-site radical hysterectomy. The fluctuation of CUSUM-T in phase 2 is probably due to change in the surgical team members. The second peak was at the 24th case. After the 24th case, the slope shifted from positive to negative and CUSUM-T decreased and reached zero. The patient’s characteristics of the first 24 cases and latter 20 cases did not differ significantly; however, the console time was significantly shorter in the latter 20 cases. Interestingly, the docking and closure times were significantly shorter in the latter 20 cases than in the first 24 cases, resulting in a shorter total operation time. This suggest that 24 cases are required for the entire surgical team, including the operator, to gain proficiency in robot-assisted single-site radical hysterectomy since the docking and closure procedures performed by the surgical team consisted of the first and second assistant and circulating nurse. The surgeon in this study was expert in MISs, including robotic surgeries. Since 2013, more than 2000 cases of robot-assisted surgery in both benign and malignant gynecologic disease were performed at our center. This factor could explain the relatively fast learning curve demonstrated in this study. Thus, the learning curve for those not familiar with robotic surgery could be different.

A recent randomized control trial reported that the DFS and overall survival of patients with cervical cancer who underwent MIS in radical hysterectomy is inferior to that in open abdominal radical hysterectomy [11,22]. Some clinicians criticized the limitations of recent randomized control trials; MIS remains as a debatable issue in cervical cancer treatment [23]. There are several retrospective studies demonstrating that the recurrence rate of open radical hysterectomy and radical hysterectomy under MIS in early cervical cancer under stage Ib1 was not significantly different [24,25,26,27]. The NCCN guideline in 2019 recommended that the patients should be carefully counseled about the short-term versus long-term outcomes and oncologic risks of the different surgical approach [7]. As a result of recent randomized controlled trials and according to NCCN guidelines, in our center informed consent about the treatment method was obtained after having sufficient discussions with patients. The survival analysis in our study consistently demonstrated that patients with stage under Ib1 showed more significantly favorable DFS than those in stage Ib2 and that stage was the only factor that influenced DFS. There was report by Li et al., which was multi-institutional study that included 1484 patients, reported that cervical cancer with tumor size less than 2 cm showed comparable oncologic outcomes between laparoscopic radical hysterectomy and open radical hysterectomy groups [28]. There are studies about optimal treatment strategies for stage Ib2 to IIa1 [29,30,31]. We believe that abdominal radical hysterectomy or CCRT are appropriate treatment strategies for stage Ib3 or IIa1 cervical cancer; however, randomized trials on patients with stage Ib2 cervical cancer are warranted in the future to establish the safety and feasibility of RSRH in patients with cervical cancer in stage Ib2.

Our study aimed to investigate the safety and feasibility of RSRH in cervical cancer. However, our study possesses the limitation of being a single-center retrospective study with a small number of cases. Based on our findings, we believe that with careful counselling, RSRH can be utilized for patients with early cervical cancer under stage Ib1; however, randomized trials for stage Ib2 or studies for larger populations are warranted. Studies including more patients with cervical cancer in stage Ib3 or IIa1 that explore the long-term clinical outcomes are also necessary.

The surgical outcomes of RSRH for early stage cervical cancer treatment were safe and acceptable. However, we cannot make a clear statement about oncologic outcomes due to the limitations of single-arm and retrospective studies. Large-scaled prospective studies are necessary in the future to validate the results of this study.

## Figures and Tables

**Figure 1 jpm-13-00733-f001:**
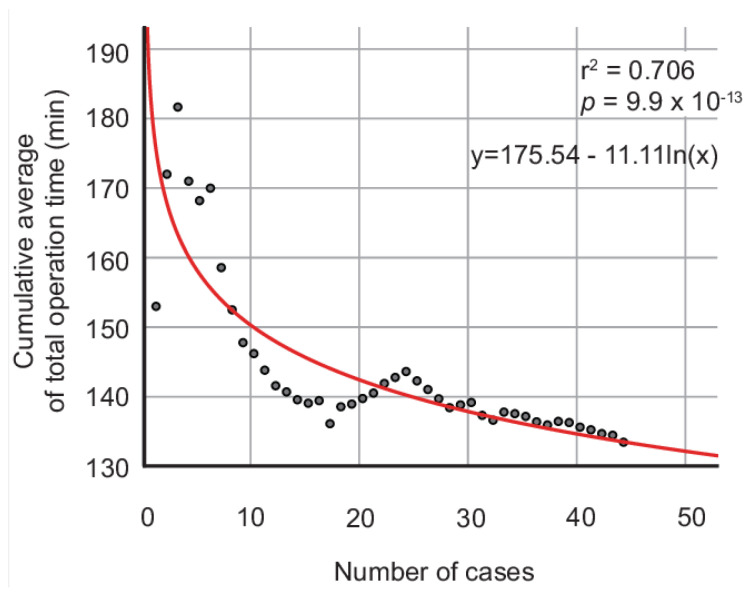
Cumulative average of total operation time of robotic single-site radical hysterectomy. Red line indicates optimized fit model of the cumulative average of the total operation time: y = 175.54 − 11.11ln(x) with optimized level r^2^ = 0.706 (*p* = 9.9 × 10^−13^).

**Figure 2 jpm-13-00733-f002:**
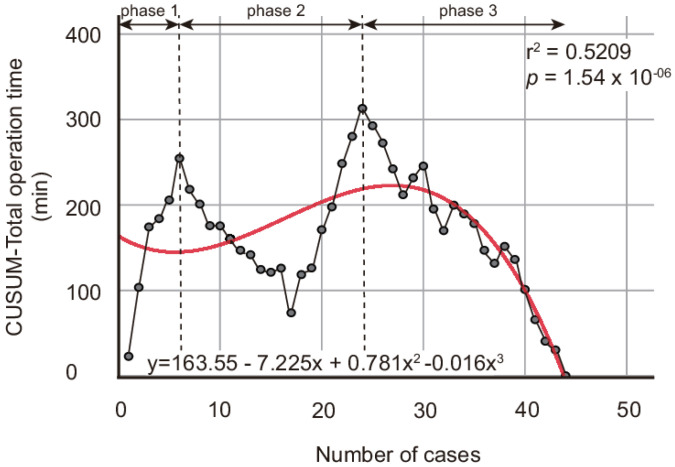
Cumulative sum (CUSUM) analysis for total operation time of robotic single-site radical hysterectomy. Red line indicates optimized fit model CUSUM-T where y = 163.55 − 7.225x + 0.781x^2^ − 0.016x^3^ with optimized level r^2^ = 0.5209 (*p* = 1.54 × 10^−6^). Two dashed lines represent breakthrough points (6th case and 24th case) and changes in phases of learning curve. Mean total operation times in phase 1, phase 2, and phase 3 were 197.7 ± 29.0, 158.5 ± 29.7, and 139.6 ± 21.25 min, respectively.

**Figure 3 jpm-13-00733-f003:**
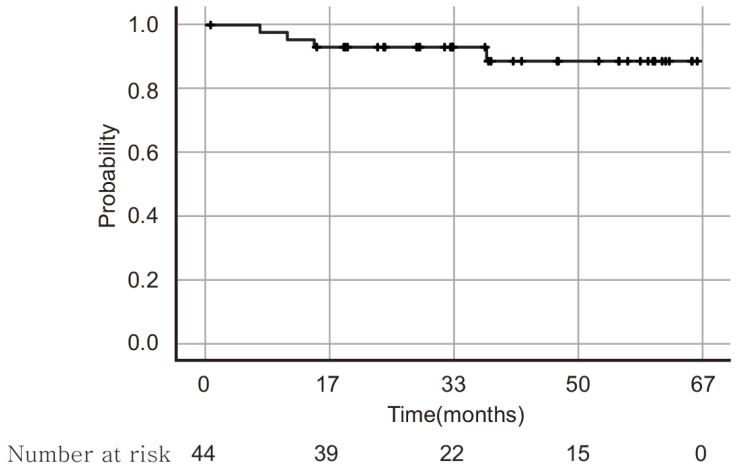
Disease-free survival rate for 44 patients who underwent robot-assisted single-site radical hysterectomy. Mean follow-up period for 44 patients was 34 months. Disease-free survival including all stages of cervical cancer at 5 years was 90.9%.

**Figure 4 jpm-13-00733-f004:**
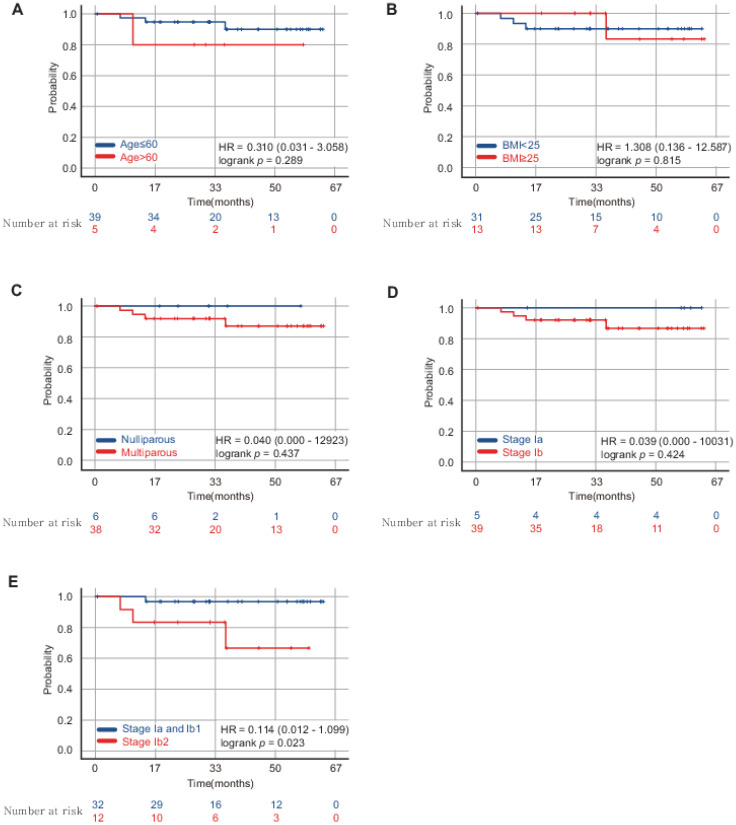
Patients with stages Ia2 and Ib1 cervical cancer showed better disease-free survival (DFS) than those with stage Ib2 cervical cancer following robotic single-site radical hysterectomy. Survival analyses were conducted through sub-dividing patients’ characteristics into age, BMI, parity, and stage. (**A**–**C**) Sub-group of patients according to the patient characteristics did not affect the DFS. (**D**) DFS was not significantly different between sub-group of five patients in stage Ia and 39 patients in stage Ib. (**E**) Stage Ia2 and Ib1 sub-group showed better DFS than stage Ib2 sub-group. HR: Hazard ratio with 95% confidence interval in brackets.

**Table 1 jpm-13-00733-t001:** Patient characteristics.

Variables	Total (n = 44)
Age {median (range), years}	45 (30–65)
Age > 60 (n, %)	5 (11.4)
BMI (mean ± SD, kg/m^2^)	23.96 ± 4.04
BMI < 18.5 (n, %)	4 (9.1)
BMI 18 to < 25 (n, %)	26 (59.1)
BMI 25 to < 30 (n, %)	10 (22.7)
BMI ≥ 30 (n, %)	4 (9.1)
Parity	
Multiparous (n, %)	38 (86.4)
Nulliparous (n, %)	6 (13.6)
Pre-operative FIGO stage	
Stage Ia1 (n, %)	0 (0.0)
Stage 1a2 (n, %)	5 (11.4)
Stage 1b1 (n, %)	27 (61.3)
Stage 1b2 (n, %)	12 (27.3)
Stage 1b3 (n, %)	0 (0.0)
Histologic type	
Squamous cell carcinoma (n, %)	33 (74.0)
Adenocarcinoma (n, %)	11 (26.0)

SD—standard deviation; BMI—body mass index; and FIGO—International Federation of Gynecology and Obstetrics.

**Table 2 jpm-13-00733-t002:** Intra- and post-operative surgical outcomes of RSRHs.

Variables	Total (n = 44)
Total operation time (mean ± SD)	156.07 ± 31.77 min
Setting time (mean ± SD)	23.49 ± 6.67 min
Preparation time (mean ± SD)	8.42 ± 4.58 min
Docking time (mean ± SD)	6.05 ± 2.85 min
Console time (mean ± SD)	95.81 ± 24.95 min
Closure time (mean ± SD)	22.30 ± 7.55 min
Conversion to laparoscopy or laparotomy (n, %)	0 (0)
Blood transfusion (n, %)	0 (0)
Retrieved lymph nodes (median [range])	9 (2–20)
Estimated blood loss (mean ± SD)	189.77 ± 132.32 mL
Complication	
Minor (n, %) ^a^	10 (22.7)
Major (n, %) ^b^	2 (4.5)
Days of hospital stay (median [range])	5 (3–60) days

^a^ Minor complications include three cases of excessive drainage, four cases of voiding difficulties, two cases of abdominal pain, and one case of free fluid collection. ^b^ Major complications include one case of vaginal cuff disruption and one case of rectal perforation. SD—standard deviation.

**Table 3 jpm-13-00733-t003:** Patient characteristics and operation time.

	N	Setting (Min)		Prep (Min)		Docking (Min)		Console (Min)		Closure (Min)		Total (Min)	
	Number	Mean	*p-*Value	Mean	*p-*Value	Mean	*p-*Value	Mean	*p-*Value	Mean	*p-*Value	Mean	*p-*Value
Age													
≤60 >60	39	23.3	0.596	8.6	0.603	6.1	0.595	94.4	0.562	21.2	0.051	153.6	0.340
	5	25.0		7.4		5.4		102.0		28.4		168.2	
BMI													
<25 ≥25	31	22.5	0.097	8.1	0.385	5.7	0.172	87.0	0.0002	21.3	0.357	144.4	0.0001
	13	26.1		9.4		6.9		115.1		23.7		181.1	
Parity													
Nulliparous Multiparous	6	30.0	0.008	8.2	0.869	5.8	0.862	94.5	0.940	15.3	0.020	153.8	0.908
	38	22.5		8.5		6.1		95.3		23.1		155.5	
Stage													
Ia2 Ib1 Ib2	32	23.1	0.488	8.0	0.318	6.2	0.457	94.5	0.767	21.3	0.302	153.1	0.483
	12	24.7		9.6		5.5		97.1		24.0		160.8	

BMI—body mass index; and prep—preparation.

**Table 4 jpm-13-00733-t004:** Surgical time of first 24 and latter 20 cases.

	First 24 Cases	Latter 20 Cases	*p-*Value
Setting time (Mean ± SD)	24.0 ± 7.0 min	22.9 ± 6.3 min	0.574
Preparation time (Mean ± SD)	8.7 ± 5.3 min	8.2 ± 3.5 min	0.738
Docking time (Mean ± SD)	7.1 ± 3.1 min	4.7 ± 1.8 min	0.003
Console time (Mean ± SD)	103.8 ± 24.8 min	85.0 ± 21.5 min	0.011
Closure time (Mean ± SD)	24.7 ± 7.8 min	18.9 ± 6.3 min	0.011
Total operation time (Mean ± SD)	168.3 ± 33.7 min	139.6 ± 21.2 min	0.002

SD—standard deviation.

**Table 5 jpm-13-00733-t005:** Oncologic outcomes.

Variables	Total (n = 44)
Tumor size	
<2 cm	32 (72.7)
≥2 cm, <3 cm	9 (20.5)
≥3 cm, <4 cm	3 (6.8)
Tumor involvement of parametrium	
No	41 (93.2)
Present	3 (6.8)
Tumor involvement of resection margin	
No	42 (95.5)
Present	2 (4.5)
Tumor involvement of lymph node	
No	44 (100)
Present	0 (0)
Adjuvant therapy (n, %)	15 (34.1)
RTx only (n, %)	2 (4.5)
CCRT (n, %)	13 (29.5)
Recurrence (n, %)	4 (9.1)
Recurred site	
Local recurrence (n)	2
Distant metastasis (n)	2
Time to recur (median, range)	16.9 (6.9–36.2) months
Cancer related death (n)	1

CCRT—chemo-radiotherapy; RTx—Radiotherapy.

## Data Availability

The data of this study are available from the corresponding author on request.

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
