# Peer review of "Robotic Single-Site Radical Hysterectomy for Early Cervical Cancer: A Single Center Experience of 5 Years"

_jpm, 2023, doi:10.3390/jpm13050733_

Round 1

Reviewer 1 Report (New Reviewer)

The authors, in a group of 44 patients with early cervical cancer in FIGO IA2-IB2 stages, evaluated 5 years of experience in the field of robotic surgery.

The authors are aware that the main disadvantage of the study is the small number of patients, which weakens the impact of the conclusions drawn.

Here are some points that I think need clarification:

Section 2.1

Line 53 - the authors write about the retrospective nature of the work, which resulted in withdrawal from obtaining consent by patients, and in line 56 - "The patients were informed about the result of resent randomized controlled trial and signed informed consent to have MIS" - in the context of the sentence in line 53 this sounds incomprehensible to me and requires comment

Section 3.1

Table 1. - more than 27% of patients had IB2 stage (meaning infiltration diameter ≥ 2 < 4 cm), and in the light of current recommendations, robotic surgery in cervical cancer should concern infiltrations up to 2 cm. Is this due to the retrospective nature of the analysis?

Section 3.4

An important aspect of the work is the analysis of the learning curve, in which the endpoints are the time of surgery and the results of the surgical technique used. They indicate that a significant shortening of the operation time occurs with the increasing number of operations performed, with two "breakthrough" points at the 6th and 24th level of the procedure. The small number of complications confirms the acquired skills. Due to the small number of patients, the number of operators performing the above-mentioned 44 procedures is an important information. How many procedures were performed by each operator?, maybe the learning curve is for one operator?

Section 3.5

As for the oncological effectiveness of the analyzed surgical technique, it is noted that some of the failures occurred after only 6.9 months (Table 5). There is no comment as to whether this early failure was local or distant, or was it a case with an infiltration diameter greater than 2 cm? Parametrium involvement was found in 3 cases. A comment would be useful if the preoperative imaging diagnostics were sufficient.

Part 4

In line 266, the authors write that over 2,000 gynecological procedures have been performed using robotic surgery since 2013, does it mean that there were only 44 cases of early cervical cancer in this group?

Author Response

Section 2.1

Line 53 - the authors write about the retrospective nature of the work, which resulted in withdrawal from obtaining consent by patients, and in line 56 - "The patients were informed about the result of resent randomized controlled trial and signed informed consent to have MIS" - in the context of the sentence in line 53 this sounds incomprehensible to me and requires comment.

Author’s response: Thank you for your kind comment. To clarify the meaning, we rewrote the sentence. As written in line 53, due to retrospective nature of the study the requirement of informed consent was waived. Our intention was to tell that the patients were counseled about the result of LACC trial and the decision of surgical approach was made on their own will.

Section 3.1

Table 1. - more than 27% of patients had IB2 stage (meaning infiltration diameter ≥ 2 < 4 cm), and in the light of current recommendations, robotic surgery in cervical cancer should concern infiltrations up to 2 cm. Is this due to the retrospective nature of the analysis?

Author’s response: Thank you for your comment. We do agree about the current recommendation. Likewise, our study showed good prognosis of stage Ia2, Ib1 cervical cancer patient included in this study but worse prognosis than that of LACC trial if Ib2 patients are included. But as you mentioned, this was retrospective study before the clear recommendation was established. The majority of stage Ib2 patients were included before LACC trial result was published. We mentioned in discussion that future randomized trials are warranted to establish the safety and feasibility of robot single site radical hysterectomy. 

Section 3.4

An important aspect of the work is the analysis of the learning curve, in which the endpoints are the time of surgery and the results of the surgical technique used. They indicate that a significant shortening of the operation time occurs with the increasing number of operations performed, with two "breakthrough" points at the 6th and 24th level of the procedure. The small number of complications confirms the acquired skills. Due to the small number of patients, the number of operators performing the above-mentioned 44 procedures is an important information. How many procedures were performed by each operator?, maybe the learning curve is for one operator?

Author’s response: We appreciate your comment. In this study, only the RSRH cases performed by one surgeon was included. To clarify, we added statement that RSRH performed in our study was performed by single surgeon.   

Section 3.5

As for the oncological effectiveness of the analyzed surgical technique, it is noted that some of the failures occurred after only 6.9 months (Table 5). There is no comment as to whether this early failure was local or distant, or was it a case with an infiltration diameter greater than 2 cm? Parametrium involvement was found in 3 cases. A comment would be useful if the preoperative imaging diagnostics were sufficient.

Author’s response: We think this is great comment. 1 patient recurred within 6.9 month without any known risk factor. The tumor size was 1.8cm and histologic subtype was squamous cell carcinoma. Without known risk factors the patient did not underwent adjuvant treatment. Tumor recurred in vaginal stump and the patient underwent CCRT and several lines of chemotherapies but unfortunately the treatment had failed. The FIGO stage according to tumor volume and lymph node status in pre-operative imaging showed consistent result after surgery but there was 3cases of unexpected parametrial involvement. As advised, we added the comment about the quality of pre-operative imaging test.

Part 4

In line 266, the authors write that over 2,000 gynecological procedures have been performed using robotic surgery since 2013, does it mean that there were only 44 cases of early cervical cancer in this group?

Author’s response: We appreciate your comment. 2,000 gynecologic surgeries included both benign and malignant diseases and both multiport. 44 cases are the only number of robot single site radical hysterectomy done by a single surgeon. There were couple more robot single site radical hysterectomy cases performed by other surgeon which was not included in this study. Including cases of laparoscopic radical hysterectomy, multiport robotic radical hysterectomy, open radical hysterectomy might be greater in number.

Reviewer 2 Report (New Reviewer)

Hello, cannot be accepted for the following reasons.  The surgical technique, the extent of parametrectomy and the technique of radical parametrectomy are not well described. A median of 9 nodes is not sufficient for systematic pelvic lymphadenectomy. Comparison with LACC is not sufficiently defended. The LACC results suggest that it is not appropriate to operate on patients from a minimally invasive approach due to the higher risk of recurrence and death (LACC/MIS group study: 91.2% 3YDFS, 93.8% 3YOS), the current study advocates a robotic approach with 5YOSS 90.9%), at the same time the current study has a high proportion of adjuvant treatment (35%) compared to LACC with 28.2% or Senticol II 14.1%.

Author Response

Hello, cannot be accepted for the following reasons.  The surgical technique, the extent of parametrectomy and the technique of radical parametrectomy are not well described. A median of 9 nodes is not sufficient for systematic pelvic lymphadenectomy. Comparison with LACC is not sufficiently defended. The LACC results suggest that it is not appropriate to operate on patients from a minimally invasive approach due to the higher risk of recurrence and death (LACC/MIS group study: 91.2% 3YDFS, 93.8% 3YOS), the current study advocates a robotic approach with 5YOSS 90.9%), at the same time the current study has a high proportion of adjuvant treatment (35%) compared to LACC with 28.2% or Senticol II 14.1%.

Author’s response: We appreciate your comment, and we will try our best to improve our manuscript. The small number of collected lymph nodes is that, we routinely perform sentinel lymph node biopsy for early cervical cancer surgery. The surgical technique is close TypeC1 hysterectomy according to Querleu and Morrow surgical classification as mentioned in the manuscript with citation. 5 Yrs OS showed RSRH including FIGO stage 1a2, 1b1, 1b2 but the 5Yrs OS of 1a2 and 1b1 patients is comparable to that of LACC trial. We totally agree about your comment and we are very discreet about making clear statement about oncologic outcomes. We admit the weakness of our study, therefore we think that future large scaled randomized controlled trials are warranted to validate the results.

Reviewer 3 Report (New Reviewer)

The article by Song et al. describes about the surgical outcomes of robotic single-site radical hysterectomy (RSRH) with respect to early cervical cancer.

Some suggestions to improve the manuscript are:

1. The figure legends in Figure 3 and Figure 4 require more explanation.

2. The title of Table 2 should be elaborated.

Author Response

  1. The figure legends in Figure 3 and Figure 4 require more explanation.

Thank you for your kind comment. As advised, we added more explanation in figure legends and in the manuscript.

  1. The title of Table 2 should be elaborated.

Thank you for your kind comment. As advised we added more explanation of table 2 in the manuscript.

Round 2

Reviewer 2 Report (New Reviewer)

Thank you for responding to my comments. They are acceptable in view of a retrospective unicentric study and thus manuscript can be accepted in this way, being aware of the current evidence and findings from the prospective studies. Congratulations!

This manuscript is a resubmission of an earlier submission. The following is a list of the peer review reports and author responses from that submission.

Round 1

Reviewer 1 Report

Introduction (line 35-37)

The standard treatment for patients from FIGO stage Ia2 to IIa1 cervical cancer who do not wish to preserve fertility is radical hysterectomy with pelvic lymph node dissection [7].

This statement is incorrect. It implies that cervical cancer stage IB3 should be treated with radical hysterectomy pus pelvic lymph node dissection. This sentence should be rewritten.

However, there were 10 cases of cervical cancer stage IB2 that undergoing only modified radical hysterectomy.

Line 38-43: this sentence was too long. It should be divided in small sentence.

The abbreviation should be considered. There were too many words of “robot assisted single-site radical hysterectomy” in this manuscript.

Line 70: what is the RUMI cuff?

Line 71-73: The following statement was incorrect.

Radical hysterectomy was sequentially performed from the dissection of both the internal and external lymph nodes in each side of the pelvis to the modified type II hysterectomy [9].

Cervical cancer stage Ia1 with lymph vascular space invasion or Ia2 was treated by modified radical hysterectomy with pelvic lymph node dissection. While cervical cancer stage IB1orIB2 was treated by radical hysterectomy with pelvic lymph node dissection.

Result part

In this study, twenty-two percent (10/44) of the cases who had histopathology report of lymph node metastasis later. These cases were classified as cervical cancer IIIC1/2P.

Ten percent (10/44) of cases had tumor involvement at parametrium or resection margin.

Thirty percent (15/44) of cases had incorrect pre-operative diagnosis. These cases should be removed from the study.

In material and method, there was no detailed of pre-operative investigation. Incorrect pre-operative diagnosis was too much.

Line 110-111: The following sentence was contrast to the Table 4.

The median count of the retrieved lymph nodes was 9 ranging from 2 to 20 110 and none of them showed tumor involvement.

From Table 4, there were 10 cases of lymph node metastasis.

Line 134: The following sentence should be adjusted as the next sentence.

The results demonstrated that the console time and the total operation time were longer in both the groups that had patients with a higher BMI.

Console and total operation time of cases with high BMI were longer than those with low BMI.

Author Response

  1. The standard treatment for patients from FIGO stage Ia2 to IIa1 cervical cancer who do not wish to preserve fertility is radical hysterectomy with pelvic lymph node dissection [7]. This statement is incorrect. It implies that cervical cancer stage IB3 should be treated with radical hysterectomy pus pelvic lymph node dissection. This sentence should be rewritten.

Author’s response: We appreciate for this important comment. What we meant by Ia2 to IIa1 cervical cancer was including Ia2, Ib1, Ib2, Ib3 and IIa1 cervical cancer. To clarify the meaning, we rewrote the sentence (See attachment).

  1. However, there were 10 cases of cervical cancer stage IB2 that undergoing only modified radical hysterectomy.

Author’s response: Type2 hysterectomy that we perform in our center is close to TypeC1 hysterectomy according to Querleu and Morrow surgical classification. The reason for using the term of type2 hysterectomy was it was not definite type3 hysterectomy. We have changed modified type 2 radical hysterectomy to type C1 radical hysterectomy according to Querleu and Morrow surgical classification and changed the citated article. 

  1. Line 38-43: this sentence was too long. It should be divided in small sentence.

Author’s response: As advised, we rewrote the sentence. (See attachment)

  1. The abbreviation should be considered. There were too many words of “robot assisted single-site radical hysterectomy” in this manuscript.

Author’s response: We appreciate this important comment. As advised, we used abbreviation for robot assisted single-site radical hysterectomy (RSRH).

  1. Line 70: what is the RUMI cuff?

Author’s response: RUMI cuff was meant by KOH cup. To clarify the meaning, we changed RUMI cuff to KOH cup. KOH cup is a vaginal fornices delinear which highlights anatomical landmarks for colpotomy during hysterectomy.

  1. Line 71-73: The following statement was incorrect.

Radical hysterectomy was sequentially performed from the dissection of both the internal and external lymph nodes in each side of the pelvis to the modified type II hysterectomy [9].

Cervical cancer stage Ia1 with lymph vascular space invasion or Ia2 was treated by modified radical hysterectomy with pelvic lymph node dissection. While cervical cancer stage IB1orIB2 was treated by radical hysterectomy with pelvic lymph node dissection.

Author’s response: We believe that the risk of parametrial invasion and LN metastasis are low in early cervical cancer. In our center, cervical cancer Ia1 with LVSI, Ia2, Ib1 and Ib2 are treated with type2 radical hysterectomy to reduce surgical morbidity and maintain oncologic outcome. Type2 hysterectomy that we perform in our center is close to TypeC1 hysterectomy according to Querleu and Morrow surgical classification. We have changed modified type 2 radical hysterectomy to type C1 radical hysterectomy according to Querleu and Morrow surgical classification.

  1. In this study, twenty-two percent (10/44) of the cases who had histopathology report of lymph node metastasis later. These cases were classified as cervical cancer IIIC1/2P.Ten percent (10/44) of cases had tumor involvement at parametrium or resection margin.Thirty percent (15/44) of cases had incorrect pre-operative diagnosis. These cases should be removed from the study.In material and method, there was no detailed of pre-operative investigation. Incorrect pre-operative diagnosis was too much.

Author’s response: We are deeply sorry about the mistake. There was no case of lymph node involvement. There was mistake in some part that the wrong version of table was inserted in the submitted manuscript. We don’t know the reason but please be generous and accept our apologies. There was 3 cases of involvement of parametrium and 2 cases of positive resection margin. This study includes patients from 2015 to 2021. Before the revised FIGO staging of cervical cancer in 2018, cervical cancer was clinically staged. We used imaging test at that time but if there is no definite parametrial invasion of tumor, it is hard to tell by image and by physical exam. We believe that these cases were pre-operatively diagnosed early cervical cancer without parametrial invasion. The stages does not change even after patients receive initial treatment. In our study we have mainly focused to demonstrate surgical outcomes of robot single-site radical hysterectomy for treatment of early cervical cancer. We have changed the conclusion of this study that surgical outcome was safe and acceptable but we cannot make clear statement about oncologic outcome. We think the cases of positive resection margin or positive parametrial invasion after surgery are necessary to show negative result of our study. It would be grateful if you reconsider this. Once again, we are deeply sorry about our mistake.   

  1. Line 110-111: The following sentence was contrast to the Table 4. The median count of the retrieved lymph nodes was 9 ranging from 2 to 20 110 and none of them showed tumor involvement. From Table 4, there were 10 cases of lymph node metastasis.

Author’s response: We are deeply sorry about the mistake. The statement of line 110-111 is correct. We have changed the table 4.  

  1. Line 134: The following sentence should be adjusted as the next sentence.

Author’s response: Thank you for the comment. As advised, we have changed the sentence.  

The changes in the manuscript is in red color.

Reviewer 2 Report

The authors present results of a small series of patients with early cervical cancer operated by a single-site radical hysterectomy at a single centre. Patients were included between 2015 until 2020. The main aim of this retrospective analysis was to assess surgical complications and oncologic outcome measured by DFS.

I have however some major concerns in regard to this manuscript. As explained in methods informed consent was waived owing to the retrospective nature of the analysis of the data. I however assume that patients signed informed consent to be operated with a new method. Especially after publication of LACC study in 2018 when level A evidence proved significant poorer oncologic outcome after minimally invasive approach in patients with early cervical cancer. The authors should clearly explain how they presented results from LACC study to patients before being operated in whether they signed informed consent. We should all be aware that LACC study was prospective randomized study with more than 600 patients. This is the most powerful evidence in medicine. Therefore puting LACC study in line with retrospective analysis, as authors did in discussion, is inappropriate. Retrospective analysis are always biased by patient selection and are always of lower quality and power to provide strong evidence. Currently, from ethical point of view, it is only acceptable to operate patients with early cervical cancer robotically, if they are included in prospective study, approved by ethical committee, signing informed consent with full knowledge of results from LACC study.

Furthermore, according to the methods and results about 44 patients, the only conclusion could be, that robotic single-site radical hysterectomy can be performed, is feasible, with acceptable acute surgical complication rate. However, the retrospective analysis of 44 patients, is far too small to make firm conclusions that robotic surgery is oncologically safe. In my opinion, authors can only report the results of DFS, but can not conclude, that the method is oncologically safe. The methods used do not allow them to make this conclusion.

Furthermore, 

Author Response

The authors present results of a small series of patients with early cervical cancer operated by a single-site radical hysterectomy at a single centre. Patients were included between 2015 until 2020. The main aim of this retrospective analysis was to assess surgical complications and oncologic outcome measured by DFS.

I have however some major concerns in regard to this manuscript. As explained in methods informed consent was waived owing to the retrospective nature of the analysis of the data. I however assume that patients signed informed consent to be operated with a new method. Especially after publication of LACC study in 2018 when level A evidence proved significant poorer oncologic outcome after minimally invasive approach in patients with early cervical cancer. The authors should clearly explain how they presented results from LACC study to patients before being operated in whether they signed informed consent. We should all be aware that LACC study was prospective randomized study with more than 600 patients. This is the most powerful evidence in medicine. Therefore putting LACC study in line with retrospective analysis, as authors did in discussion, is inappropriate. Retrospective analysis are always biased by patient selection and are always of lower quality and power to provide strong evidence. Currently, from ethical point of view, it is only acceptable to operate patients with early cervical cancer robotically, if they are included in prospective study, approved by ethical committee, signing informed consent with full knowledge of results from LACC study.

Furthermore, according to the methods and results about 44 patients, the only conclusion could be, that robotic single-site radical hysterectomy can be performed, is feasible, with acceptable acute surgical complication rate. However, the retrospective analysis of 44 patients, is far too small to make firm conclusions that robotic surgery is oncologically safe. In my opinion, authors can only report the results of DFS, but can not conclude, that the method is oncologically safe. The methods used do not allow them to make this conclusion.

Author’s response: Thank you for this important comment. In our institution, cervical cancer patient with tumor size less than 2cm in pre-operation exam usually undergo radical surgery with MIS. Since LACC trial was published in 2018, the results of LACC trial were informed during the patient counselling. Patients who agree to undergo MIS, radical surgery with multiport or single-site assisted by either robot or laparoscopy was done. The surgical method is clearly explained and is written in informed consent of surgery. As advised, we have added this comment in discussion. Since this study was not designed from the beginning, the informed consent about the enrollment to this retrospective study was not provided at the time. As this study only used medical records of patients, the informed consent about the retrospective study was waived off and is approved by IRB in our institution. We do respect the result of LACC trial, and after it was published we recommend MIS only to patients with small volume of cervical cancer. We appreciate your comments and as advised, we clarified the nature of studies about Open Vs. MIS in cervical cancer in discussion and changed the order. We changed the conclusion as advised. (See attachment)

The changes of manuscript is in red color.

Round 2

Reviewer 1 Report

From previous comment, the author recruited ten cases of cervical cancer IIIC1P into the study. I already suggest to remove these cases from the study. The authors claimed that only the mistake of writing. There was no case with positive lymph node metastasis. However from old version, there were 15 cases who underwent adjuvant treatment by radiation. After correction, there were still 10 cases who had undergone radiation without appropriate explanation.

Author Response

Dear reviewer. We appreciate your comments and we sincerely respect your insights and opinions. We tried our best to follow your comments in the previous review as well as this time. Thank you for pointing out the errors and mistakes. We have rewrote the sentence in the manuscript that you have indicated. We would be grateful if you can review once again. Thank you. 

In our center, all the cancer patients are discussed in tumor board meeting. Cervical cancer patients who underwent radical hysterectomy are reviewed after the surgery with gynecologic oncologists, pathologists and radiation oncologists. Mostly based on NCCN guideline and GOG-092 experts discuss and decides plan for the patients.

To clarify this and to follow your comment, we rewrote the sentence. Please see attachment written in red color.

Thank you and best regards    

Reviewer 2 Report

Thank you to the authors to respond to the comments. If the manuscript will be published I would suggest to include information about counceling the patients in the Metarials nad methods section. Under the title ''Patients'' I would suggest to add that patients were informed about the results of LACC study and signed informed consent to have MIS. I believe it is crucial to put that in the materials and methods. Thank you.  

Author Response

We appreciate for this important comment. As advised, we rewrote the method and materials. Please see the attachment written in red color.
